# Novel Ubiquitin Specific Protease-13 Inhibitors Alleviate Neurodegenerative Pathology

**DOI:** 10.3390/metabo11090622

**Published:** 2021-09-15

**Authors:** Xiaoguang Liu, Kaluvu Balaraman, Ciarán C. Lynch, Michaeline Hebron, Christian Wolf, Charbel Moussa

**Affiliations:** 1Department of Neurology, Translational Neurotherapeutics Program, Laboratory for Dementia and Parkinsonism, Lewy Body Dementia Association, Research Center of Excellence, Georgetown University Medical Center, 4000 Reservoir Rd. NW, Building D, Room 203-C, Washington, DC 20007-2145, USA; xl371@georgetown.edu (X.L.); mlh88@georgetown.edu (M.H.); 2Department of Chemistry, Georgetown University & Medicinal Chemistry Shared Resource, Georgetown University Medical Center, Washington, DC 20057, USA; bk562@georgetown.edu (K.B.); ccl86@georgetown.edu (C.C.L.); cw27@georgetown.edu (C.W.)

**Keywords:** neurodegeneration, Ubiquitin Specific Protease-13 (USP13), USP13 inhibitors

## Abstract

Ubiquitin Specific Protease-13 (USP13) promotes protein de-ubiquitination and is poorly understood in neurodegeneration. USP13 is upregulated in Alzheimer’s disease (AD) and Parkinson’s disease (PD), and USP13 knockdown via shRNA reduces neurotoxic proteins and increases proteasome activity in models of neurodegeneration. We synthesized novel analogues of spautin-1 which is a non-specific USP13 inhibitor but unable to penetrate the brain. Our synthesized small molecule compounds are able to enter the brain, more potently inhibit USP13, and significantly reduce alpha-synuclein levels in vivo and in vitro. USP13 inhibition in transgenic mutant alpha-synuclein (A53T) mice increased the ubiquitination of alpha-synuclein and reduced its protein levels. The data suggest that novel USP13 inhibitors improve neurodegenerative pathology via antagonism of de-ubiquitination, thus alleviating neurotoxic protein burden in neurodegenerative diseases.

## 1. Introduction

Ubiquitin Specific Protease (USP)-13, also named ubiquitin carboxyl-terminal hydrolase 13 which belongs to the family of cysteine proteases [1,2,3], promotes cleavage of ubiquitin chains, resulting in protein de-ubiquitination [4]. USP13 is widely distributed in normal human tissue, including the cytosol and nucleoplasm [4,5]. Neurodegenerative diseases, including Dementia with Lewy Bodies (DLB) and Parkinson’s disease (PD), are characterized by accumulation of misfolded alpha-synuclein, which forms Lewy body (LB) inclusions [6,7]. Alzheimer’s disease (AD) is characterized by tau hyper-phosphorylation that forms inclusions called tangles, along with plaque deposits containing amyloid-β (Aβ) peptides [8,9]. These neurotoxic proteins (tau, Aβ and alpha-synuclein) may co-exist in AD, PD and DLB [10,11,12,13,14,15,16,17,18]. USP13 protein expression levels are elevated in post-mortem cortex of AD and the nigrostriatum of PD [19,20]. USP13 knockdown via shRNA regulates the E3 ubiquitin ligase activity of autosomal recessive PD-linked parkin and increases the ubiquitination of alpha-synuclein and hyper-phosphorylated tau (p-tau), leading to enhanced proteasome clearance [20,21]. USP13 also regulates de-ubiquitination of stress granules [22]. Therefore, there is an urgent need to develop USP13 inhibitors and investigate their pharmacology in neurodegeneration.

Spautin-1 is currently a known specific inhibitor to USP10 and USP13 with IC_50_s of 0.6–0.7 µM [23]. It inhibits autophagy by deregulating the formation of VPS34 complex acting over USP13 and USP10 that modifies Beclin-1 ubiquitination [23]. However, it has very poor penetration to brain. There are very limited studies about spautin-1 in neurodegeneration. It was mainly used as a small molecular tool to study autophagy in cancer research. For instance, spautin-1 enhances imatinib-induced apoptosis in chronic myeloid leukemia [24]. Spautin-1 ameliorates acute pancreatitis via inhibiting impaired autophagy and alleviating calcium overload [25]. 

To further elucidate the role of USP13 in neurodegeneration, we developed a small library of specific small molecule inhibitors of USP13. We demonstrate that brain-penetrant USP13 inhibitors regulate (de)-ubiquitination of neurotoxic proteins and may alleviate neurodegenerative pathology. Our previous findings and the current data suggest that USP13 inhibition prevents protein de-ubiquitination, thus providing a feasible neurotherapeutic target to facilitate misfolded protein clearance. The novel library of small molecules constitutes new tools for further investigation to assess the therapeutic potential of USP13 inhibition in neurodegeneration.

## 2. Results

### 2.1. Chemical Synthesis and Cell Viability Assays

Spautin-1 does not cross the blood–brain barrier (BBB) and non-specifically inhibits USP13 and USP10 and affects proteasome and autophagy pathways [23]. Therefore, we developed analogues of spautin-1 that are more potent inhibitors of USP13 and are small molecules that can enter the brain. USP-13 inhibitors bearing 6-fluoroquinoline, thieno [3,2-*b*] pyridine and 3-nitrocoumarin backbones were synthesized as shown in Figure 1. Nucleophilic aromatic substitution reactions using aniline (2) with either 7-chlorothieno [3,2-*b*]pyridine (1) or 4-chloro-6-fluoroquinoline (3) in DMSO gave BK50118-A and BK50118-B in 98% and 95% yield, respectively. Similarly, BK50118-C was obtained from 4-chloro-6-fluoroquinoline (3) and benzylalcohol (4) in the presence of NaH in 97% yield. Palladium catalyzed C-N bond formation produced CL3-499 from 4-chloro-6-fluoroquinoline (3) and benzylamine (5) in 81% yield. The amino acid derivatives CL3-512 and CL3-514 were synthesized from 4-chloro-3-nitrocoumarin (6) and the methyl esters of *L*-phenylalanine and *L*-tyrosine in 90% and 42% yield, respectively, in the presence of K_2_CO_3_ at room temperature. The products were purified by column chromatography and characterized by ^1^H, ^13^C and ^19^F NMR spectroscopy as well as high resolution mass spectrometry as described in more detail in Appendix A. We thus generated a small library of six compounds (Figure 2A).

Cell viability using lactate dehydrogenase (LDH) and 3-(4,5-dimethylthiazol-2-yl)-2,5-diphenyltetrazolium bromide (MTT) assays in SH-SY5Y human neuroblastoma cells indicated that BK50118-A (Figure 2B) decreased cell viability near and above 0.1 μM compared to DMSO or untreated controls. BK50118-B and BK50118-C (Figure 2C,D) decreased cell viability near and above 100 μM compared to DMSO or untreated controls. CL3-499 (Figure 2G) decreased cell viability near and above 1 mM compared to DMSO or untreated controls. CL3-512 and CL3-514 (Figure 2E,F) did not show cell toxicity at the range of concentrations studied compared to DMSO or untreated controls.

### 2.2. Novel Small Molecules Are Potent USP13 Inhibitors

We next determined whether the novel compounds inhibit USP13 activities and clear toxic proteins. Stably transfected human neuroblastoma SH-SY5Y cells with human wild-type alpha-synuclein were treated with 1 mM, 100 µM, 10 µM, 1 µM, 0.1 µM, 0.01 µM and 0.001 µM of each of BK50118-A, BK50118-B, BK50118-C, CL3-499, CL3-512 and CL3-514 dissolved in an equivalent volume of DMSO for 5 h. USP13 activity via functional ELISA showed that all of the 6 compounds inhibited USP13 (Figure 3A–F) with maximal inhibition coefficient (IC_50_) ranging from 0.11 nM to 2.13 nM (Figure 3G). Measurement of human alpha-synuclein via ELISA showed that BK50118-C (Figure 3J) most potently decreased alpha-synuclein levels, reaching almost complete alpha-synuclein clearance at 1 mM and 100 µM. The other 5 compounds were also able to lower alpha-synuclein levels but with less potency (Figure 3H,I,K–M).

### 2.3. Absorption, Distribution, Metabolism and Excretion (ADME) Studies

We focused on BK50118-C as the most potent USP13 inhibitor to determine the pharmacokinetics (PK) of this molecule. Wild type C57BL6 mice were injected with 10 mg/kg BK50118-C versus DMSO, and the brain and serum were isolated at 0, 1, 2, 4, 6, 8 and 12 h; extracted in water and examined by mass spectrometry. The concentration of BK50118-C peaked at 1 h (T_max_) both in the serum and brain (Table 1), and a maximal concentration (C_max_) of 81.49 nM and 354.63 nM were reached in the brain and serum, respectively. The bioavailabilities of the drug (AUC, area under the curve) were 164.3 nM×hand 599.4 nM×h in the brain and serum, respectively. Elimination (T_1/2_) was 2.32 h for brain and 1.84 h in serum. The ratio of serum: brain reached 28%, indicating that this drug abundantly enters the brain.

### 2.4. BK50118-C Reduces Alpha-Synuclein, Increases Alpha-Synuclein Ubiquitination and Improves Neuronal Survival in Mice

Transgenic A53T mice harbor the arginine to threonine (A53T) mutation of human alpha-synuclein under the control of prion promoter and have abundant alpha-synuclein in the striatum as early as 3 months of age [26].

Male and female TgA53T mice (15 months old) were treated daily with intraperitoneal injection of DMSO versus 10 mg/kg or 40 mg/kg BK50118-C for 7 days. WB of midbrain lysates showed that human alpha-synuclein was significantly reduced at the above dosages (Figure 4A,B 1st blot). Immunoprecipitation of alpha-synuclein protein from midbrain extracts followed by ubiquitin WB (Figure 4B, 2nd blot) showed increased ubiquitination in 40 mg/kg BK50118-C compared to DMSO and 10 mg/kg BK50118-C. Immunoprecipitation of ubiquitin from midbrain extracts followed by alpha-synuclein WB (Figure 4B, 3nd blot) showed no monomeric alpha-synuclein but increased ubiquitination of alpha-synuclein in 40 mg/kg BK50118-C compared to DMSO and 10 mg/kg BK50118-C. ELISA measurement of alpha-synuclein in the same extracts (Figure 4C) confirmed that human alpha-synuclein was reduced by about 30% (10 mg/kg) and 56% (40 mg/kg), respectively (Figure 4C).

TgA53T mice also demonstrated an elevated state of tauopathy in striata, suggesting that tauopathy is a common feature of synucleinopathies [27]. Therefore, we also measured tau levels. There was no effect on murine tau levels (Figure 4D).

Immunostaining of 20µm thick brain sections showed human alpha-synuclein staining in both cortex (Figure 4E,G,I) and striatum (Figure 4J,L,N) in DMSO treated mice. BK50118-C treatment (40 mg/Kg) significantly decreased human alpha-synuclein staining in both the cortex (Figure 4F,H,I) and striatum (Figure 4K,M,I) compared to DMSO. Quantification showed that BK50118-C (40 mg/Kg) significantly reduced the number of human alpha-synuclein positive neurons by 42% in cortex and 40% in striatum (Figure 4I,N).

Nissl staining showed that BK50118-C significantly increased neuron counts in cortex (Appendix A), striatum (Appendix A) and substantia nigra (SN) (Appendix A) compared to DMSO (Appendix A), as verified by quantification of Nissl+ cells in cortex (Appendix A), striatum (Appendix A) and SN (Appendix A).

### 2.5. BK50118-C Increases Alpha-Synuclein Ubiquitination and Has Minimal Effects on Tyrosine Hydroxylase Levels in Striatum of TgA53T Mice

To ascertain the mechanistic effects of BK50118-C on alpha-synuclein clearance, we performed immunostaining in striatum of TgA53T mice. Human alpha-synuclein staining showed the level of human alpha-synuclein in DMSO (Figure 5A) compared to BK50118-C (40 mg/Kg) treated mice (Figure 5B). Ubiquitin staining in DMSO treated striatum (Figure 5C) showed the level of ubiquitin compared to mice treated with 40 mg/Kg of BK50118-C (Figure 5D). Merged alpha-synuclein and ubiquitin staining in DMSO (Figure 5E), compared to BK50118-C (Figure 5F) showed that alpha-synuclein co-localized with ubiquitin in mice treated with BK50118-C. Optic density of alpha-synuclein-ubiquitin staining showed a significant increase (130%) in co-localization in the striatum when mice were injected with BK50118-C compared to DMSO (Figure 5G). We also performed the staining of tyrosine hydroxylase (TH) in the striatum of TgA53T mice. TH is the enzyme responsible for catalyzing the rate limiting step in synthesis of L-3,4-dihydroxyphenylalanine(L-DOPA) which is a precursor for dopamine (DA). Staining of TH will help to evaluate the status of DA producing neurons in the SN and their terminals in the striatum. Staining of tyrosine hydroxylase (TH) in the striatum of TgA53T mice did not show any noticeable effects in BK50118-C (Figure 5I,K,M,N) compared to DMSO treated mice (Figure 5H,J,L,N).

## 3. Discussion

The current research demonstrates that pharmacological inhibition of USP13 lowers neurotoxic protein levels, including alpha-synuclein and improves neurodegenerative pathology. We previously showed that shRNAs which target USP13 expression can reduce alpha-synuclein via autophagy and or the proteasome in vivo and in vitro [20,21]. The current results are in agreement with our previous findings that knockdown via shRNA and pharmacological inhibition of USP13 increase clearance of ubiquitinited proteins [20,21]. We synthesized novel small molecules and potent inhibitors of USP13 that effectively reduce neurotoxic protein levels and protect against neuronal death. Mechanistically, these novel USP13 inhibitors mitigate the detrimental activity of USP13 that de-ubiquitinates alpha-synuclein. USP13 inhibition may antagonize de-ubiquitination of neurotoxic proteins and prevent their aggregation into LBs and tangles via promotion of ubiquitination to facilitate autophagy and proteasome protein clearance [20,21]. Collectively these findings suggest that USP13 is a therapeutic target in neurodegenerative diseases [28].

USP13 appears to play a critical role in the regulation of protein clearance mechanisms. USP13 interacts with the Beclin-1-parkin protein complex and controls autophagosome maturation [19,29,30]. We previously demonstrated that USP13 reduces the E3 ubiquitin ligase activity of parkin via de-ubiquitination [21], while USP13 knockdown increases parkin ubiquitination and activity [21]. USP13 may also be critical for the regulation of other E3 ubiquitin ligases and autophagy. The class III phosphoinositide 3-kinase (PIK3C3) and vacuolar protein sorting 34 (VPS34) form a complex that regulates autophagy initiation and progression [31]. The E3 ubiquitin ligase NEDD4/NEDD4-1 activates PIK3C3 via ubiquitination, and USP13 decreases ubiquitination of PIK3C3 and reduces autophagy [31,32]. Knockout of either NEDD4-1 or USP13 increases ubiquitination and degradation of VPS34, thus attenuating autophagosome formation [31]. The ubiquitin-recognition protein Ufd1 facilitates endoplasmic reticulum (ER)-associated degradation (ERAD) of misfolded proteins via USP13 [33]. Downregulation of USP13 expression reduces PTEN [34].

The novel USP inhibitors show very high potency to inhibit USP13, and to our knowledge, they represent the first library of small molecule, brain penetrant USP13 inhibitors. Spautin-1 does not cross the BBB, and it inhibits both USP10 and USP13 and affects the proteasome and autophagy at much higher concentration than the novel USP13 library [23]. Our novel USP13 inhibitors are non-toxic at the concentrations used, but more work is needed to determine their activity against other USPs. Therefore, further investigation to determine the specificity of the novel molecules to other USPs and additional in vivo studies will be performed to determine PK parameters of a wider range of drug concentrations and their safety and efficacy. Our future investigation will focus on mechanistic aspects beyond de-ubiquitination to include specific mechanisms of protein clearance via autophagy and the proteasome in USP13 knockout mice.

Many other de-ubiquitinases (DUBs), including USP14 [35] and Ovarian Tumor (OTU, Ubiquitin Aldehyde Binding-1) OTUB-1 [36] regulate tau (de)ubiquitination and control its proteasome clearance. USP8 regulates alpha-synuclein (de)ubiquitination and proteasome clearance [37]. USP19 is implicated in polyglutamine-expanded (polyQ) diseases, including Huntington disease (HD) and spinocerebellar ataxia [38,39,40]. USP20 and USP30 control of the fusion of amphisomes and endosomes with lysosomes to form auto-lysosomes [3]. USP33 de-ubiquitinates parkin and antagonizes its role in mitophagy [41]. PTEN-induced kinase 1 (PINK1) activates parkin to degrade depolarized mitochondria [42,43,44] and promotes misfolded protein clearance [44].

## 4. Materials and Methods

### 4.1. Transgenic Mice, Stereotaxic Surgery and Treatment

Transgenic TgA53T mice were used in the experiment. rTg4510 mice are hemizygous and have the Tet-responsive element as well as a mouse prion protein promoter sequence directing expression of the human P301L tau. Hemizygous rTg4510 mice are bred to CaMKIIa-tTA mice resulting in the rTg4510 bi-transgenic model with conditional P301L tau expression in the forebrain. Transgenic A53T mice harbor the arginine to threonine (A53T) mutation of human alpha-synuclein under the control of prion promoter and have abundant alpha-synuclein in the striatum as early as 3 months of age [26].

BK50118-C is one of novel small molecule inhibitors of USP13 synthesized in our laboratory. For BK50118-C treatment, approximately 15 months old (±1 month) male and female TgA53T mice were treated with intraperitoneal injection of vehicle (DMSO) or BK50118-C at the daily dosage of 10 mg/Kg or 40 mg/Kg for 7 days. Animals were age and sex-matched in their respective treatment cohorts. All animals were sacrificed after treatment. All experiments were conducted in full compliance with the recommendations of Georgetown University Animal Care and Use Committee. These experiments were conducted under animal protocol No. 2016-1194 titled, Misfolded Protein Clearance in Neurodegeneration, which was approved by the Georgetown University IACUC. Protocol Information: IACUC Protocol Number: 2016-1194. Protocol Approval Date: 14 September 2020 Expiration Date: 14 September 2023.

### 4.2. Chemical Synthesis of USP13 Inhibitors

Commercially available 7-chlorothieno[3,2-*b*]pyridine (1), aniline (2), 4-chloro-6-fluoroquinoline (3), benzyl alcohol (4), benzyl amine (5), 4-chloro-3-nitro-2*H*-chromen-2-one (6), reagents and solvents were used as purchased without further purification. Methyl (*S*)-phenylalaninate (7) and methyl (*S*)-tyrosinate (8) were obtained from their HCl salts by means of extraction with aqueous NaHCO_3_ and EtOAc. NMR spectra were obtained at 400 MHz (^1^H NMR), 100 MHz (^13^C NMR) and 376 MHz (^19^F NMR) in deuterated solvents. Reaction products were purified by column chromatography on silica gel (particle size 40–63 μm) as described below.

For the synthesis of BK50118-A, BK50118-B and BK50118-C (Appendix A), an 8 mL pressure vessel was charged with a heteroaryl chloride and two equivalents of either aniline (2) or benzyl alcohol (4) in DMSO unless noted otherwise. The pressure vessel was then placed in a 100 °C oil bath, and the mixture was stirred for 24 h. After full conversion was achieved based on ^1^H NMR analysis, the reaction mixture was quenched with NaHCO_3_ and extracted with EtOAc. The combined organic layers were extracted with water and dried over sodium sulfate, and the solvent was removed in vacuo. The crude product was purified by flash chromatography on silica gel using with hexanes-ethyl acetate as mobile phase as described below.

Compound BK50118-A (*N*-phenylthieno[3,2-*b*]pyridin-7-amine) was obtained as a colorless solid in 98% yield (133 mg, 0.59 mmol) from 7-chlorothieno[3,2-*b*]pyridine (1) (101 mg, 0.6 mmol) and aniline (2) (111 mg, 1.2 mmol) in 1 mL of DMSO after 24 h at 100 °C by following the general procedure described above. The crude product was purified using flash chromatography on silica gel using hexanes/EtOAc (1:1) as mobile phase. *R_f_* = 0.3 (hexanes/EtOAc, 1:1); ^1^H NMR (400 MHz, Chloroform-*d*) δ = 8.42 (d, *J* = 5.4 Hz, 1H), 7.63 (d, *J* = 5.4 Hz, 1H), 7.53 (d, *J* = 5.4 Hz, 1H), 7.45–7.37 (m, 2H), 7.32–7.25 (m, 2H), 7.19 (m, 1H), 6.92 (d, *J* = 5.4 Hz, 1H), 6.12 (s, 1H); ^13^C NMR (100 MHz, Chloroform-*d*) δ = 157.8, 148.9, 146.3, 139.4, 129.7, 128.1, 126.5, 124.9, 122.6, 120.8, 102.3. HRMS (ESI-TOF) m/z: [M+H]^+^ calcd for C_13_H_11_N_2_S 227.0637, found 227.0639.

Compound BK50118-B (6-fluoro-*N*-phenylquinolin-4-amine) was obtained as a colorless solid in 95% yield (135 mg, 0.57 mmol) from 4-chloro-6-fluoroquinoline (3) (109 mg, 0.6 mmol) and aniline (2) (111 mg, 1.2 mmol) in 1 mL of DMSO after 24 h at 100 °C by following the general procedure described above. The crude product was purified using flash chromatography on silica gel using hexanes/EtOAc (1:1) as mobile phase. *R_f_* = 0.3 (hexanes/EtOAc, 1:1); ^1^H NMR (400 MHz, Chloroform-*d*) δ = 8.54 (d, *J* = 5.2 Hz, 1H), 8.04 (dd, *J* = 9.9, 5.2 Hz, 1H), 7.58 (dd, *J* = 9.9, 2.7 Hz, 1H), 7.49–7.37 (m, 3H), 7.28 (d, *J* = 7.7 Hz, 2H), 7.19 (m, 1H), 7.01 (d, *J* = 5.2 Hz, 1H), 6.59 (s, 1H); ^13^C NMR (100 MHz, Chloroform-*d*) δ = 160.2 (d, *J* = 246.7 Hz), 150.3 (d, *J* = 2.3 Hz), 147.3 (d, *J* = 5.0 Hz), 146.4, 139.9, 132.8 (d, *J* = 9.0 Hz), 129.9, 124.9, 122.7, 120.5 (d, *J* = 8.5 Hz), 119.4 (d, *J* = 25.1 Hz), 104.1 (d, *J* = 23.1 Hz), 103.0; ^19^F NMR (376 MHz, Chloroform-*d*) δ = -114.04 (m). HRMS (ESI-TOF) m/z: [M+H]^+^ calcd for C_15_H_12_FN_2_ 239.0979, found 239.0980.

Compound BK50118-C (4-benzyloxy-6-fluoroquinoline) was obtained as a colorless solid in 97% yield (147 mg, 0.58 mmol) from 4-chloro-6-fluoroquinoline (3) (109 mg, 0.6 mmol), benzyl alcohol (4) (130 mg, 1.2 mmol) and NaH (0.6 mmol) in 2 mL of THF after 16 h at 100 °C by following the general procedure described above. The crude product was purified using flash chromatography on silica gel using hexanes/EtOAc (1:1) as mobile phase. *R_f_* = 0.5 (hexanes/EtOAc, 1:1); ^1^H NMR (400 MHz, Chloroform-*d*) δ = 8.70 (d, *J* = 5.2 Hz, 1H), 8.03 (dd, *J* = 9.5, 5.2 Hz, 1H), 7.85 (dd, *J* = 9.5, 2.9 Hz, 1H), 7.54–7.33 (m, 6H), 6.80 (d, *J* = 5.2 Hz, 1H), 5.27 (s, 2H); ^13^C NMR (100 MHz, Chloroform-*d*) δ = 160.8 (d, *J* = 5.1 Hz), 160.2 (d, *J* = 246.7 Hz), 150.6 (d, *J* = 2.5 Hz), 146.4 (d, *J* = 1.2 Hz), 135.5, 131.5 (d, *J* = 9.0 Hz), 128.9, 128.6, 127.5, 122.2 (d, *J* = 9.6 Hz), 119.9 (d, *J* = 25.6 Hz), 105.9 (d, *J* = 23.4 Hz), 101.7, 70.5; ^19^F NMR (376 MHz, Chloroform-*d*) δ = −113.82 (m). HRMS (ESI-TOF) m/z: [M+H]^+^ calcd for C_16_H_13_FNO 254.0976, found 254.0978.

For the synthesis of CL3-499 (Appendix A), an 8 mL sealed vial was charged with 4-chloro-6-fluoroquinoline (3) (100 mg, 0.55 mmol), Pd_2_(dba)_3_ (10.1 mg, 0.01 mmol, 2 mol%), 1,3-bis(2,4,6-trimethylphenyl)imidazolium chloride (TMPCl) (11.3 mg, 0.03 mmol, 6 mol%), NaO*t*-Bu (169.2 mg, 1.76 mmol), benzylamine (5) (177 mg, 1.65 mmol) and dioxane (1.5 mL). The vial was then placed in a 100 °C oil bath and stirred for 56 h. After cooling to room temperature, the reaction mixture was quenched with aqueous NaHCO_3_ and extracted with EtOAc. The combined organic layers were dried over sodium sulfate, and the solvent was removed in vacuo. The crude product was purified by flash chromatography on silica gel using hexanes-EtOAc (8:2) as mobile phase and recrystallized from hexanes-EtOAc (1:1).

Compound CL3-499 (*N*-benzyl-6-fluoroquinolin-4-amine) was obtained as a yellow solid in 81% yield (112 mg, 0.45 mmol). *R_f_* = 0.16 (hexanes/EtOAc, 1:1); ^1^H NMR (400 MHz, Chloroform-*d*) δ 8.51 (d, *J* = 5.2 Hz, 1H), 7.99 (dd, *J* = 5.8 Hz, *J* = 5.3 Hz, 1H), 7.43–7.31 (m, 7H), 6.47 (d, *J* = 5.2 Hz, 1H), 5.18 (bs, 1H), 4.53 (d, *J* = 5.2 Hz, 2H); ^13^C NMR (100 MHz, Chloroform-*d*) δ 159.7 (d, *J* = 245.0 Hz), 150.4 (d, *J* = 2.3 Hz), 149.0 (d, *J* = 4.7 Hz), 145.5, 137.3, 132.4 (d, *J* = 8.9 Hz), 128.9, 127.9, 127.5, 119.2 (d, *J* = 8.6 Hz), 118.8 (d, *J* = 24.9 Hz), 103.7 (d, *J* = 22.6 Hz), 99.8, 47.6; ^19^F NMR (376 MHz, Chloroform-*d*) δ −114.97 (m). HRMS (ESI-TOF) m/z: [M+H]^+^ calcd for C_16_H_13_FN_2_ 253.1222, found 253.1140.

For the synthesis of CL3-512 and CL3-514 (Appendix A), an 8 mL vial was charged with 4-chloro-3-nitro-2*H*-chromen-2-one (6), amino ester derivative 7 or 8, K_2_CO_3_ and ACN:H_2_O (6.8 mL, 2.4:1 *v*/*v*). The reaction mixture was stirred at 25 °C for 2 h. After full conversion was achieved based on TLC analysis, the pH was adjusted to 7, and the reaction mixture was extracted with EtOAc. The combined organic layers were washed with water and dried over sodium sulfate, and the solvent was removed in vacuo. The crude products were purified by flash chromatography on silica gel as described below.

Compound CL3-512 ((*S*)-methyl (3-nitro-2-oxo-2*H*-chromen-4-yl)-phenylalaninate) was obtained after column purification using DCM/MeOH (9:1) as mobile phase as a colorless solid in 90% yield (274 mg, 0.8 mmol) from 4-chloro-3-nitro-2H-chromen-2-one (**6**) (200 mg, 0.89 mmol), methyl (*S*)-phenylalaninate (**7**) (146.5 mg, 0.89 mmol) and K_2_CO_3_ (246 mg, 1.78 mmol) by following the general procedure described above. *R_f_* = 0.44 (DCM/MeOH, 9:1); ^1^H NMR (400 MHz, Methanol-*d4*) δ 7.98 (d, *J* = 8.3 Hz, 1H), 7.67 (dd, *J* = 9.1 Hz, 7.2 Hz, 1H), 7.37 (dd, *J* = 9.0 Hz, 7.3 Hz, 1H), 7.28 (d, *J* = 8.3 Hz, 1H), 7.21–7.13 (m, 5H), 4.57 (bs, 1H), 3.34 (dd, *J* = 14.1 Hz, 4.5 Hz, 1H), 3.29 (s, 3H), 3.22 (dd, *J* = 14.1 Hz, 8.7 Hz, 1H); ^13^C NMR (100 MHz, Methanol-*d4*) δ 172.1, 155.9, 151.0, 135.8, 133.8, 128.7, 128.7, 128.3, 126.9, 124.5, 123.5, 117.2, 116.7, 113.1, 58.8, 53.4, 38.1. HRMS (ESI-TOF) m/z: [M+H]^+^ calcd for C_19_H_16_N_2_O_6_ 369.1087, found 369.1084.

Compound CL3-514 ((*S*)-methyl (3-nitro-2-oxo-2*H*-chromen-4-yl)-tyrosinate) was obtained after column purification using DCM/MeOH (9:1) as mobile phase as a colorless solid in 42% yield (137.1 mg, 0.37 mmol) from 4-chloro-3-nitro-2*H*-chromen-2-one (6) (200 mg, 0.89 mmol), methyl (*S*)-tyrosinate (8) (161 mg, 0.89 mmol) and K_2_CO_3_ (246 mg, 1.78 mmol) by following the general procedure described above. *R_f_* = 0.21 (DCM/MeOH, 9:1); ^1^H NMR (400 MHz, Methanol-*d4*) δ 7.95 (dd, *J* = 8.2, 1.4 Hz, 1H), 7.61 (dd, *J* = 8.2, 1.3 Hz, 1H), 7.34 (dd, *J* = 8.3, 1.3 Hz, 1H), 7.22 (dd, *J* = 8.3, 1.4 Hz, 1H), 6.95 (d, *J* = 8.2 Hz, 2H), 6.61 (d, *J* = 8.2 Hz, 2H), 4.49 (bs, 1H), 3.77 (s, 3H), 3.21 (dd, *J* = 14.2, 4.9 Hz, 1H), 3.09 (dd, *J* = 14.2, 8.7 Hz, 1H). ^13^C NMR (100 MHz, Methanol-*d4*) δ 170.6, 156.4, 155.9, 151.4, 134.0, 129.8, 125.9, 124.5, 123.8, 117.4, 115.1, 113.3, 59.4, 52.0, 37.7. HRMS (ESI-TOF) m/z: [M+H]^+^ calcd for C_19_H_16_N_2_O_7_ 385.1036, found 385.1033.

### 4.3. Cell LINES, Transfection and Treatment

Human SH-SY5Y neuroblastoma cells were procured from the Tissue Culture and Biobanking Shared Resource at Georgetown University Lombardi Comprehensive Cancer Center. Stably transfected SH-SY5Y cells expressing human wild-type alpha-synuclein were developed via selective resistance to G418 disulfate salt (Sigma-Aldrich, A1720, St. Louis, MO, USA). Cell identity was not further authenticated. SH-SY5Y cells were cultured in DMEM with Ham’s F12 (1:1) (ThermoFisher, 11765054, Waltham, MA, USA) with 10% FBS, 1% PenStrep and 1% L-glutamine. Cells were plated at a density tailored to reach 70–80% confluence at the beginning of every experiment.

SH-SY5Y cells were transiently transfected for 24 h with human wild-type α-synuclein using Fugene HD transfection reagent (Promega, E2311, Madison, WI, USA) according to manufacturer’s protocol. Cells were treated with 1 mM, 0.1 mM, 0.01 mM, 1 µM, 0.1 µM, 0.01 µM and 0.001 µM of each novel compounds (BK50118-A, BK50118-B, BK50118-C, CL3-499, CL3-512 and CL3-514) dissolved in DMSO for 5 h. Cells were then harvested on ice by removing culture medium and adding 0.2 mL 1x sodium-tris, EDTA, NP-40 (STEN) buffer (50 mM Tris (pH 7.6), 150 mM NaCl, 2 mM EDTA, 0.2 % NP-40, 0.2 % with Halt protease and phosphatase inhibitor solution (ThermoFisher, 78446). Cells were detached with a cell scraper and collected into centrifuge tubes and incubated at 4 °C for 30 min with agitation. Samples were stored at −80 °C and used for ELISA analyses. For cell viability assay, SHSY-5Y cells stably expressing human wild-type alpha-synuclein were treated with 1 mM, 0.1 mM, 0.01 mM, 1 µM, 0.1 µM, 0.01 µM and 0.001 µM of each novel compound (BK50118-A, BK50118-B, BK50118-C, CL3-499, CL3-512 and CL3-514) dissolved in DMSO for 5 h. Cell viability was determined via lactate dehydrogenase assay on culture media (Thermofisher, 88954) and MTT assay on plated cells (Thermofisher, V13154).

### 4.4. Western Blot Analysis

To extract the soluble proteins from mouse midbrain lysates, tissues were isolated and homogenized in 1x STEN buffer (50 mM Tris (pH 7.6), 150 mM NaCl, 2 mM EDTA, 0.2% NP-40, 0.2% BSA, 20 mMPMSF and protease cocktail inhibitor), centrifuged at 10,000× *g* for 20 min at 4 °C, and the supernatant containing the soluble protein fraction was collected. Extracts were analyzed by Western blot (WB) on 4–12% SDS NuPAGE Bis-Tris gel (Invitrogen, NP0301BOX). Beta-actin (β-actin) was probed (1:3000) with monoclonal antibody (Emdmillipore, MAB1501R, Burlington, MA, USA). Human alpha-synuclein was probed (1:2000) with monoclonal antibody (Thermo Fisher, AHB0261, Rockford, IL, USA). USP13 was probed (1:1000) with polyclonal antibody (ThermoFisher, PA5-12014, Rockford, IL, USA). Ubiquitin was probed (1:5000) with polyclonal antibody (Thermo Fisher, PA3-16717, Rockford, IL, USA). WBs were quantified by densitometry using Quantity One 4.6.3 software (Bio Rad, Hercules, CA, USA) and Image J.

### 4.5. Enzyme-Linked Immunosorbent ASSAY (ELISA)

The ELISA for USP13 (MyBioSource, Cat # MBS9335287, San Diego, CA, USA), Human alpha-synuclein (Biolegend, Cat # 844101, San Diego, CA, USA) and specific p-Tau ser396 (Invitrogen, KHB7031, Waltham, MA, USA) were performed on cell or brain tissue extracts as described above according to manufacturers’ protocol.

### 4.6. Immunoprecipitation (IP)

Mouse brain tissues were homogenized in 1x STEN buffer, and the soluble fraction was isolated as indicated above. The lysates were pre-cleaned with immobilized recombinant protein A/G agarose (Santa Crutz, sc-2003, Dallas, TX, USA) and centrifuged at 2500× *g* for 1 min at 4 °C. The supernatant was recovered and quantified by protein assay, and a total of 300 µg protein was incubated overnight at 4 °C with primary anti-alpha-synuclein (1:200, Thermofisher, AHB0261) mouse antibodies or anti-ubiquitin (1:100) (Thermo Fisher, PA3–16717, Rockford, IL, USA) antibodies in the presence of sepharose G and an IgG control with primary antibodies. The immunoprecipitates were collected by centrifugation at 2500× *g* for 3 min at 4 °C, washed 5× in PBS, with spins of 3 min, 2500× *g* using detergent-free buffer for the last washing step, and the proteins were eluted according to Pierce instructions (Pierce #20365, Rockford, IL, USA). After IP, the samples were size-fractionated on 4–12% SDS-NuPAGE and transferred onto 0.45 µm nitrocellulose membranes. WB detection was then performed using horseradish peroxidase (HRP)-conjugated secondary antibodies.

### 4.7. Immunohistology

Animals were deeply anesthetized with a mixture of xylazine and ketamine (1:8), washed with normal saline for 1 min and then perfused with 4% paraformaldehyde (PFA) for 15–20 min. Brains were quickly dissected out and immediately stored in 4% PFA for 24 h at 4 °C and then transferred to 30% sucrose at 4 °C for 48 h. Brains were cut using a cryostat microtome into 20 μm thick coronal sections and stored at −20 °C.

Immunohistochemistry was performed on the 20 µm thick brain sections for evaluation of ubiquitination of alpha-synuclein. The antibodies used were human alpha-synuclein monoclonal antibody (Thermo Fisher, AHB0261, Rockford, IL, USA) and ubiquitin polyclonal antibody (Thermo Fisher, PA3-16717, Rockford, IL, USA). The optic densitometry of co-localization of ubiquitin with alpha-synuclein was measured using Image J. Tyrosine hydroxylase (TH) is the limiting enzyme in DA synthesis, so probing for TH+ neurons will help to evaluate the status of DA producing neurons in the SN and their terminals in the striatum. We performed fluorescent staining of TH+ in striatum and conducted optic densitometry measurement of striatal DA terminals. Nuclear staining with 4, 6-diamidino-2-phenylindole was performed according to manufacturer’s protocols (Life Technologies, Rockford, IL, USA).

DAB staining was performed for alpha-synuclein (Thermo Fisher, AHB0261, Rockford, IL, USA) on the 20 µm thick mouse brain sections, and stereological counting of alpha-synuclein+ neurons counterstained with Nissl was conducted.

### 4.8. Nissl and Silver Staining

Nissl staining was performed using FD Cresyl Violet Solution™ Regular Strength (FD NeuroTechnologies, Cat PS102-01, Columbia, MD, USA) as per manufacturer’s instructions. Silver staining was performed using FD NeuroSilver™ (FD NeuroTechnologies, Cat PK301,) as per manufacturer’s instructions.

### 4.9. Pharmacokinetics Studies

C57BL/6 mice received a single intraperitoneal dose (10 mg/kg) of BK50118-C. Brain and serum samples were collected at 1, 2, 3, 4, 8 and 12 h (n = 3 per time point). Animals injected with vehicle (DMSO) were used for background subtraction. Stock solution of the drug (1 mg/mL) and internal standards were prepared in methanol. Intermediate solutions used for the calibrators and control samples were serial-diluted in methanol/water (1:1). Preparation of the calibration curve standards and quality samples (QC) were performed by mixing the intermediate dilutions in blank samples (brain homogenates, serum). The internal standard working solution contained deuterium labeled BK50118-C-d7 at the concentration of 5 ng/mL diluted in acetonitrile (ACN)/ethyl acetate (4:1). Serum and brain samples were stored at −80 °C and then thawed to room temperature prior to preparation. The brains were homogenized in MilliQ water (1 mg brain: 10 µL water). Proteins were precipitated in both brain and serum samples by mixing 25 µL aqueous sample with 75 µL internal standard working solution. The mixture was centrifuged at 12,300× *g* for 5 min. Thereafter, 75 µL of each supernatant and 25 µL of MilliQ water were pipetted into a 96-well PCR plate (Fisher Scientific, Dawsonville, GA, USA).

The concentrations of BK50118-C in the brain tissue and serum samples were measured by ultrahigh performance liquid chromatography tandem mass spectrometry (UHPLC-MS/MS). Briefly, the UHPLC-MS/MS system included an Elute HTG binary gradient UHPLC pump, an Elute column oven and an EVOQ Elite triple quadrupole mass spectrometer (all from Bruker Daltonik GmbH, Bremen, Germany) equipped with an electrospray ionization (ESI) source operating in a positive mode. The samples were injected by use of a PAL auto sampler (CTC Analytics, Zwingen, Switzerland) equipped with a 10-µL sample loop; the samples were kept in a PAL stack cooler for 6 microtiter plates and operating at +6 °C. The system was controlled by a Compass 2.0/HyStar 4.0 software (Bruker); the compound screening and quantitation was performed by a TASQ 2.2 data acquisition and processing software (Bruker). The mass spectrometer was supplied by nitrogen and air generated by a Genius 3045 nitrogen/air generator (Peak Scientific Instruments, Inchinnan, Scotland, UK). The ESI parameters were as follows: probe gas flow 50, nebulizer gas flow 60, probe temperature +400 °C, cone gas flow 20, cone temperature +350 °C, CID gas Ar 1.5 mTorr. The mass spectra were scanned in the MRM mode to find the optimal collision energies for the test compounds and their respective precursor ions.

For chromatographic separation, an YMC-Ultra HT Hydrosphere C18 column (2.0 × 100 mm, 2 µm particle size) and an YMC-Hydrosphere 2.1 × 5 mm guard column (YMC Co. Ltd., Kyoto, Japan) were used. The mobile phase A was 10 mM ammonium formate, pH 4.3, in water; the mobile phase B was 10 mM ammonium formate, pH 4.3, in 10% water, 50% acetonitrile and 40% isopropyl alcohol. The mobile phase gradient was as follows (min—A/B%): 0 min—50/50, 0.5 min—50/50, 2.0 min—20/80, 3 min—20/80, 3.1 min—50/50, 4.5 min—50/50. The flow rate was 450 µL/min, and the column temperature was set at 50 °C. The sample injection volume was 10 µL.

### 4.10. Statistical and Data Analysis

All statistical analyses were performed using GraphPad Prism version 5.0 (GraphPad software, Inc, San Diego, CA, USA). Two-tailed student’s *t*-test was used in comparison of means of two groups. Ordinary one-way analysis of variance (ANOVA) followed by Newman–Keuls or Dunnett comparison post hoc tests were used in comparison of means of multiple groups. Asterisks denote actual *p*-value significances (* < 0.05, ** < 0.01, *** < 0.001 and **** < 0.0001), and N is the number of animals or the number of independent experiments (cell culture) per group. Unless otherwise indicated, data are expressed as Mean ± SD. IC50 to USP13 levels was also estimated using GraphPad Prism version 5.0 (GraphPad software, Inc, San Diego, CA, USA).

## 5. Conclusions

In conclusion, these findings suggest that USP13 inhibition may oppose protein aggregation and inclusion formation. The role of USP13 in regulating the ubiquitination and de-ubiquitination cycle and initiation and control of autophagy are integral to cell homeostasis and survival. USP13 inhibition via our novel analogues may provide balance of ubiquitination and de-ubiquitination for toxic protein degradation in neurodegenerative diseases.

## Figures and Tables

**Figure 1 metabolites-11-00622-f001:**
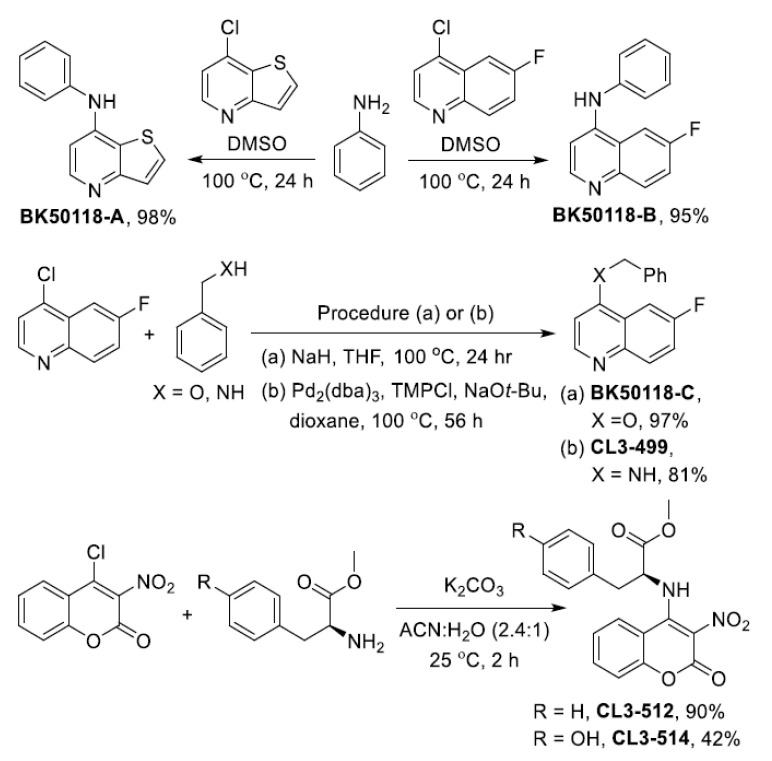
Scheme of synthesis of spautin-1 analogues.

**Figure 2 metabolites-11-00622-f002:**
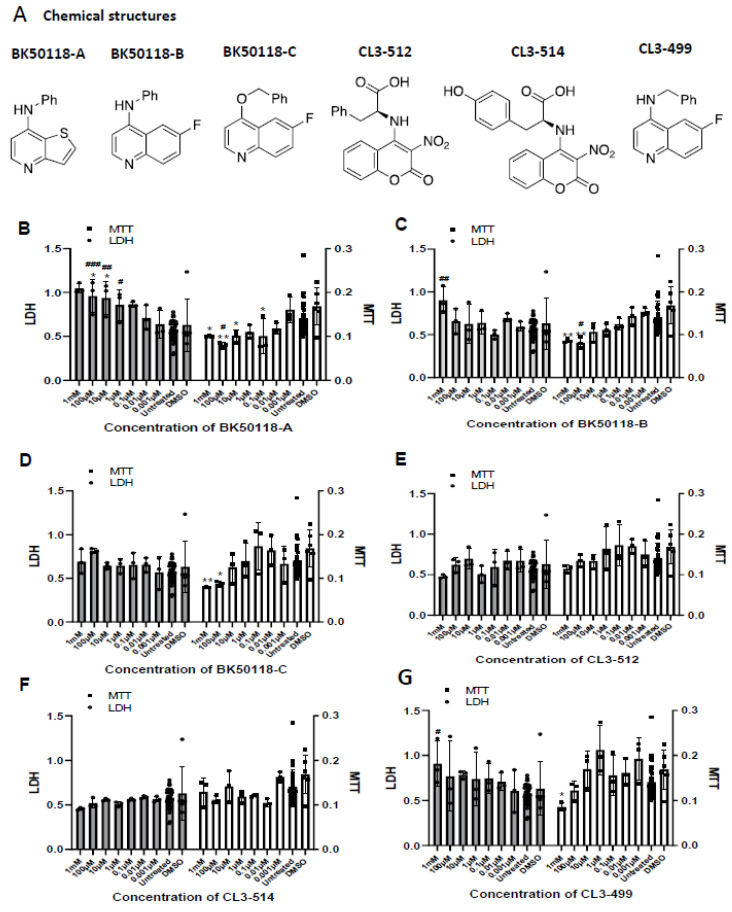
Chemical structures and cell viability assays of the 6 novel small molecule compounds in vitro. (**A**) Chemical structures of BK50118-A, BK50118-B, BK50118-C, CL3-512, CL3-514 and CL3-499. (**B**–**G**) LDH and MTT Cell viability assays of the 6 small molecule compounds. SH-SY5Y cells were treated with a serial of concentrations of each individual compound for 5 h. * *p* < 0.05, ** *p* < 0.01 to DMSO group, ^#^ *p* < 0.05, ^##^ *p* < 0.01, ^###^ *p* < 0.001 to untreated group; ordinary one-way ANOVA; N = 3–30 per treatment group.

**Figure 3 metabolites-11-00622-f003:**
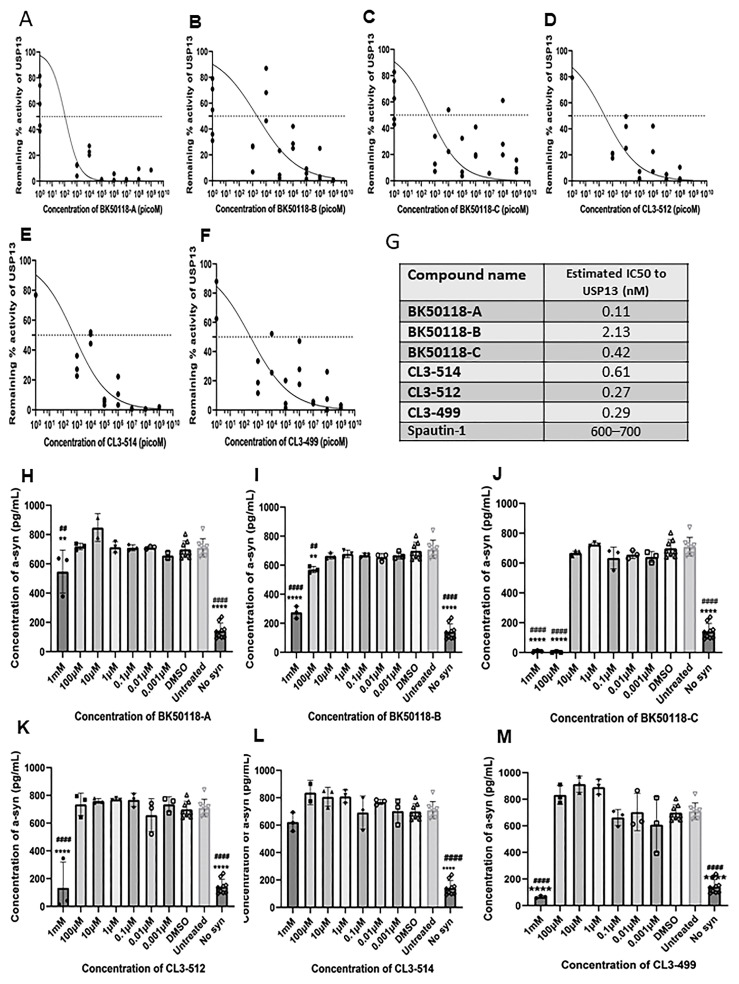
The effects of the novel small molecule compounds on the USP13 activities and alpha-synuclein in vitro. The SH-SY5Y cells were transinfected with human alpha-synuclein plasmid. Cells were treated with a serial of concentrations of each individual compound for 5 h. ELISA was performed for (**A**–**F**) USP13, and (**G**) estimated IC50 to USP13 was calculated. ELISA for (**H**–**M**) alpha-synuclein. ** *p* < 0.01, **** *p* < 0.0001 to DMSO group, ^##^ *p* < 0.01, ^####^ *p* < 0.0001 to untreated group; ordinary one-way ANOVA. N = 3–9 per treatment group.

**Figure 4 metabolites-11-00622-f004:**
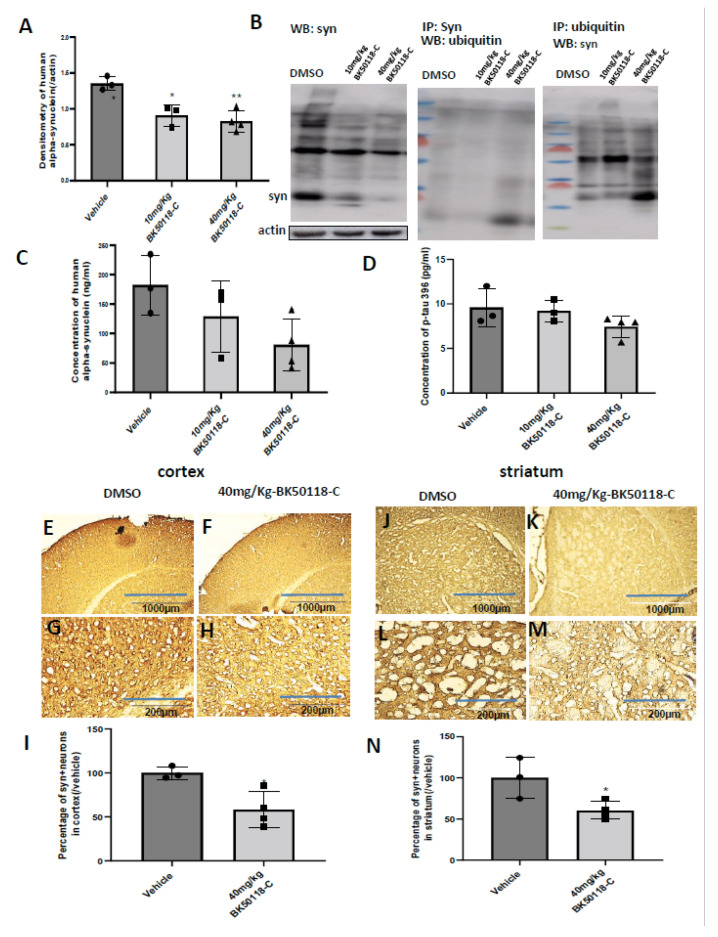
BK50118-C significantly reduced alpha-synuclein but not p-tau levels in TgA53T mice. Male and female TgA53T mice were treated with intraperitoneal injection of vehicle or BK50118-C at the daily dosage of 10 mg/Kg or 40 mg/Kg for 7 days. WB of midbrain lysates showing (**A**) the levels of alpha-synuclein relative to actin on 4–12% SDS-NuPAGE gel in the above mice. (**B**) The first blot is the WB densitometry. The 2nd and 3rd blots are IP alpha-synuclein (syn) or ubiquitin. ELISA showing (**C**) the levels of alpha-synuclein and (**D**) ptau396 following BK50118-C treatment. Asterisks indicate statistic significant difference * *p* < 0.05 or ** *p* < 0.01 vs. corresponding vehicle. Ordinary one-way ANOVA was used for the analysis. N = 3–4 mice per group. DAB +Nissl staining. C significantly reduced alpha-synuclein levels in cortex (**F**/**H**) and striatum (**K**/**M**) compared to the corresponding vehicles (**E**/**G**) or (**J**/**L**)**,** verified by the quantification of alpha-synuclein positive cells in cortex (**I**) and striatum (**N**). Two-tailed student t test was used for analysis, and * indicates *p* < 0.05 vs. vehicle. N = 3–4 per group. All values presented as Mean ± SD. Scale bars: 100 µm (**E**,**F**,**J**,**K**) or 200 µm (**G**,**H**,**L**,**M**).

**Figure 5 metabolites-11-00622-f005:**
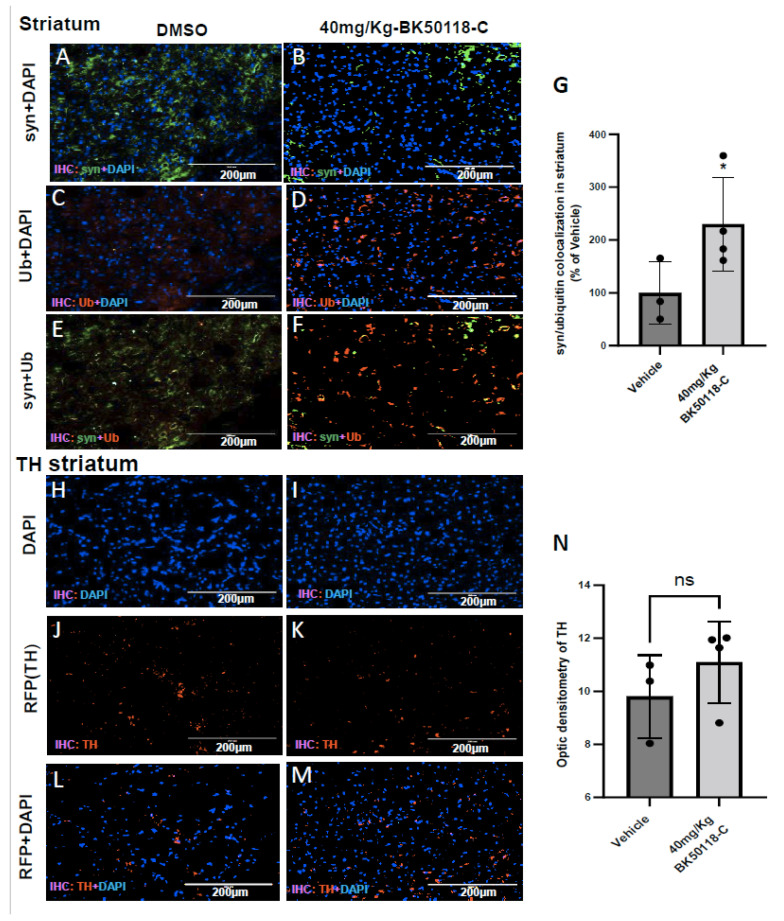
BK50118-C significantly increased alpha-synuclein ubiquitination but had minimal effects on tyrosine hydroxylase (TH) levels in striatum of TgA53T mice. Male and female TgA53T mice were treated with intraperitoneal injection of vehicle or BK50118-C at the daily dosage of 40 mg/Kg for 7 days. Immunochemistry assay of 20 µm thick brain sections showed alpha-synuclein (green), ubiquitination (red) and DAPI (blue) staining in the striatum of TgA53T mice treated with either DMSO (**A**,**C**,**E**) or BK50118-C (**B**,**D**,**F**). BK50118-C increased alpha-synuclein ubiquitination, verified by optic density of co-localization (**G**). Immunostaining showed TH (red) and DAPI (blue) staining in vehicle DMSO (**H**,**J**,**L**) and BK50118-C (**I**,**K**,**M**). BK50118-C had minimal effects on TH levels in striatum, verified by quantification of optic density (**N**). Asterisk indicates statistic significant difference * *p* < 0.05; two-tailed student’s t test was used for analysis. N = 3–4 mice per group. All values presented as Mean ± SD. Scale bars: 200 µm.

**Table 1 metabolites-11-00622-t001:** Pharmacological parameters of BK50118-C measured in wild type mice (C57B6) injected with 10 mg/kg of deuterated BK50118-C and tissues were collected over 12 h after injection.

BK50118-C	Dosage (mg/kg, I.P)	10.00
	Total drug injected (nmol)	1185.77
Brain	C_max_ (nM)	81.49 ± 69.97
	C_max_ (ng/mL)	20.62 ± 17.70
	T_max_ (h)	1.00
	AUC (nM. h)	164.3 ± 39.49
	T_1/2_ (elimination) (h)	2.32
Serum	Dosage (mg/Kg)	10.00
	C_max_ (nM)	354.63 ± 272
	C_max_ (ng/mL)	89.72 ± 68.81
	T_max_ (h)	1.00
	AUC (nM. h)	599.4 ± 142.9
	T_1/2_ (elimination) (h)	1.84
	Ratio of Serum/Brain (%)	28.0%

## Data Availability

This study includes no data deposited in external repositories.

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
