# Peer review of "Novel Ubiquitin Specific Protease-13 Inhibitors Alleviate Neurodegenerative Pathology"

_metabolites, 2021, doi:10.3390/metabo11090622_

Round 1

Reviewer 1 Report

The authors synthesized novel brain penetrable Ubiquitin Specific Protease-13 inhibitors and verified their effectiveness on inhibiting USP13 and regulating alpha-synuclein ubiquitination and protein level in vitro and in vivo. These novel drugs thus have potential values in research as a USP13 inhibitor and therapeutic application for neurodegenerative disorders. The study was well organized and clearly written. The only minor issue is the specificity of the novel inhibitors against USP13 was not tested in the study, as the authors had pointed out in the discussion. Therefore, it is better to mention the novel inhibitors directly rather than USP13 inhibition in conclusion since the effects observed in vivo may also come from inhibition of other USPs in addition to USP13.

Author Response

we thank the reviewer fo this comment. We agree and we reworded and consolidated the discussion as follows:

The novel USP inhibitors show very high potency to inhibit USP13 and to our knowledge they represent the first library of small molecule, brain penetrant USP13 inhibitors. Spautin-1 does not cross the BBB and it inhibits both USP10 and USP13 and affects the proteasome and autophagy at much higher concentration than the novel USP13 library [24]. Our novel USP13 inhibitors are non-toxic at the concentrations used, but more work is needed to determine their activity against other USPs. Therefore, further investigation to determine the specificity of the novel molecules to other USPs, and additional in vivo studies will be performed to determine PK parameters of a wider range of drug concentrations and their safety and efficacy. Our future investigation will focus on mechanistic aspects beyond de-ubiquitination to include specific mechanisms of protein clearance via autophagy and the proteasome in USP13 knockout mice.

Reviewer 2 Report

In this manuscript, Lui et al try to identify novel USP13 inhibitors that can improve neurodegenerative pathology.  In general, the manuscript is well written and proper statistical analysis is performed. I have below comments that would improve the manuscript further.

Major;

Cell viability assays indicate toxicity at or above 100uM concentration yet the author shows that the same concentrations reduce alpha-synuclein levels significantly figure H-M. These observations dampen the claim of the suitability of compounds in vivo studies.

What is the rationale for choosing a 10mg/kg dose and 40mg/kg?

Replace all bar graphs with scatter plots.

All figures should be replaced with high-quality figures. e.g. in fig 2 and 3 x-axis labels are barely readable. 

Minor;

reference 19 is not relevant to this study. 

plate well format used in vitro assays is missing from the methods section.

add full western blot images to the supplementary material.

conclusion: protein aggregation and inclusion formation assays are not reported in this study therefore this statement should be re-written.

Author Response

Cell viability assays indicate toxicity at or above 100uM concentration yet the author shows that the same concentrations reduce alpha-synuclein levels significantly figure H-M. These observations dampen the claim of the suitability of compounds in vivo studies.

What is the rationale for choosing a 10mg/kg dose and 40mg/kg?

The reviewer is right that there is a discrepancy between MTT and LDH assays in cell culture. This is necessary to show here as the differences in the assays is something to be acknowledged when looking at toxicity in culture models. The reduction of alpha-synuclein level especially with BK50118A was encouraging. Translation of the concentration of the drug in vitro to mg/kg in mice is difficult and often it is inaccurate. Therefore, we performed a viability experiment using an ascending dose of the drug in mice and reached a safe concentration of 80mg/kg before we observed any signs of in vivo toxicity (lethargy, scruffiness, or even death etc..) for at least 3 months and performed silver staining to ascertain that no cell death was observed in vivo. Therefore, the 10mg/kg and 40mg/kg were empirically determined.

We added scatter plots to bar graphs.

We replaced with high-quality figures. 

Minor;

We reference 19.

plate well format used in vitro assays is in the methods section.

We added the full WBs.

conclusion: protein aggregation and inclusion formation statement was re-written.

Reviewer 3 Report

In this manuscript Liu et al. has synthesized and characterized 6 chemical compounds as potential inhibitors of USP13. They have tested the effect of these compounds on cell viability and on alpha-synuclein levels and demonstrated the effect of one of these compounds (BK501118-C) on brain tissue of TgA53T mice. This is an interesting study and the compound tested in mice shows promise for further characterization.

Some specific comments that can improve the manuscript is as follows:

  • More background information on USP13 biology and literature on existing inhibitors (like Spautin) and how these compare to them should be included.

  • How specific are these compounds for USP13? Were they tested against other USPs?
  • USP13 ELISA (in figure 3) is not a direct measure of USP13 activity in presence of the inhibitor. Use of Ub-AMC or probes to measure Ubiquitin-USP13 binding is a more appropriate measure of USP13 activity. Also, replicates in this ELISA have high deviation.
  • Alpha-synuclein levels in Figure 3 should be supported by alpha-synuclein ubiquitination assays to check whether use of USP13 inhibitors increases ubiquitinated alpha-synuclein in cultured cells.
  • Some background information on TgA53T mice model and measurement of p-Tau levels should be provided in the text in the results section. Also the rationale for tyrosine hydroxylase staining should be included in the text.
  • 40mg/kg drug does not seem to affect alpha-synuclein levels in the western blot in Fig 4.
  • The effect of the inhibitor on Ubiquitin staining in Fig 5 is quite pronounced. Can this be due to inhibition of multiple USPs?

Minor points:

  • X-axis labels of Fig 3 A-F is differently formatted
  • Some scale bars in IHC images in Fig 4 is not clear.

Author Response

Some specific comments that can improve the manuscript is as follows:

Comment: More background information on USP13 biology and literature on existing inhibitors (like Spautin) and how these compare to them should be included.

  • Response: added in “introduction”
  •  

Comment: How specific are these compounds for USP13? Were they tested against other USPs?

  • Response: We have not tested against other USPs. However, the IC50 of BK50118-C to USP13 is about 0.42 nM which is lower than the known specific USP13 inhibitor spautin-1 (600-700 nM). This suggests that BK50118-C is a very potent USP13 inhibitor, and probably specific to USP13. We will do further studies to test its specificity to USP13 and other USPs as discussed in the manuscript.
  •  

Comment: USP13 ELISA (in figure 3) is not a direct measure of USP13 activity in presence of the inhibitor. Use of Ub-AMC or probes to measure Ubiquitin-USP13 binding is a more appropriate measure of USP13 activity. Also, replicates in this ELISA have high deviation.

  • Response: Thanks for your comments. We have changed the Y-axis label to “USP13 level” rather than “activity”. We agree that USP13 ELISA (Figure 3) is not a direct measure of USP13 activity, but the binding of USP13 functional structure to its antibody in ELISA could reflect the activity.  In “raw” tissue samples, it will be more appropriate and specific to use ELISA to measure USP13 instead of use of Ub-AMC or probes since Ub-AMC will be non-specific.

Comment: Alpha-synuclein levels in Figure 3 should be supported by alpha-synuclein ubiquitination assays to check whether use of USP13 inhibitors increases ubiquitinated alpha-synuclein in cultured cells.

  • Response: We have done immunoprecipitation (IP) experiment to check whether use of USP13 inhibitors increases ubiquitinated alpha-synuclein in TgA53T mice. We added the result to the test and Figure 4.

Comment: Some background information on TgA53T mice model and measurement of p-Tau levels should be provided in the text in the results section. Also the rationale for tyrosine hydroxylase staining should be included in the text.

  • Response: done

Comment: 40mg/kg drug does not seem to affect alpha-synuclein levels in the western blot in Fig 4.

  • Response: We repeated the Westen Blots and selected performed analysis to confirm previous results.

Comment: The effect of the inhibitor on Ubiquitin staining in Fig 5 is quite pronounced. Can this be due to inhibition of multiple USPs?

  • Response: As we mentioned in discussion, we will further test the effects of compounds on other USPs. Yes, there is a possibility that other USPs are involved.

Minor points:

Comments: X-axis labels of Fig 3 A-F is differently formatted

  • Response: Have uniformed all the X-axis

Some scale bars in IHC images in Fig 4 is not clear.

  • Response: fixed the scale bars

Round 2

Reviewer 2 Report

The authors have addressed my comments.

Reviewer 3 Report

The authors have addressed most of the concerns of this reviewer either by performing experiments or altering the text.  I have no further queries.